# Sleep Quality, Mental Health and Learning among High School Students after Reopening Schools during the COVID-19 Pandemic: Results of a Cross-Sectional Online Survey

**DOI:** 10.3390/ijerph19052553

**Published:** 2022-02-23

**Authors:** Kristijonas Puteikis, Ainė Mameniškytė, Rūta Mameniškienė

**Affiliations:** 1Faculty of Medicine, Vilnius University, 03101 Vilnius, Lithuania; kristijonas.puteikis@mf.stud.vu.lt; 2Vilnius Žirmūnai Gymnasium, 09106 Vilnius, Lithuania; aine.mameniskyte@gmail.com; 3Centre for Neurology, Faculty of Medicine, Vilnius University, 08661 Vilnius, Lithuania

**Keywords:** adolescents, anxiety, physical health, survey

## Abstract

We aimed to assess whether high school students’ sleep quality, mental health and learning changed during the COVID-19 pandemic as adolescents transitioned from learning online back to studying in person. We conducted an anonymous online cross-sectional survey study at three competitive high schools in Vilnius, Lithuania, after they were reopened. Students provided subjective views on their study quality, their health as well as daily life while studying either virtually or in person and completed the Beck depression inventory (BDI), the Generalized anxiety scale-7 (GAD-7) and the Pittsburgh sleep quality index (PSQI). Among 628 (70.4% female) respondents, 268 (42.7%), 342 (54.5%) and 398 (63.4%) are suspected to have depression, an anxiety disorder or poor sleep, respectively. Students reported better study quality (Z = −12.435, *p* < 0.001) and physical health (Z = −9.176, *p* < 0.001), but worse sleep quality (Z = −19.489, *p* < 0.001), shorter sleep duration (Z = −19.509, *p* < 0.001) and worse self-reported mental health (Z = −2.220, *p* < 0.05) while learning in person. However, higher scores of in-person study quality and physical health were associated with lower depression and anxiety levels as well as better sleep. Our study suggests that the reopening of schools may exacerbate sleep and mental health-related issues among high school students but also be beneficial for their academic development and levels of physical activity.

## 1. Introduction

After the worldwide spread of SARS-CoV-2 was declared a pandemic in March 2020, many countries employed nationwide lockdowns to prevent the collapse of their medical systems [1]. One of the restrictive measures enforced during lockdowns was school closures. They were thought to limit the transmission of SARS-CoV-2 among children and adolescents and prevent them from infecting relatives at home [2]. It remains debatable whether this practice helps to effectively decrease the spread of SARS-CoV-2 and if school closures are acceptable at all, considering the benefits schools provide as both educational and societal institutions [3,4,5]. After transitioning to online learning, students faced a completely different daily routine that changed their learning experience, sleep patterns and social interactions [6,7,8]. While it has been argued that lockdowns may disrupt sleep, exacerbate mental health problems and reduce physical student activity, highschoolers could also benefit from later waking hours and flexible learning schedules [9,10,11,12,13,14,15,16]. However, there remains little data on how high school students perceive the return to live education after several months of online learning [17,18,19,20,21].

In recent years, the view of health being a state of “complete well-being” shifted towards a dynamic definition of health, which revolves around the capacity to “react to all kinds of environmental events having the desired emotional, cognitive, and behavioral responses and avoiding those undesirable ones” [22]. Similarly, recent conceptualizations of mental health emphasize that it is “a dynamic state of internal equilibrium which enables individuals to use their abilities in harmony with universal values of society” [23]. Self-rated single-item measures of overall mental health or the evaluation of scales used to detect selected clinical entities (e.g., anxiety, depression or poor quality of sleep) can help to quantify the benefits and disadvantages that the COVID-19 pandemic had for the health of the youngest generations of individuals in our societies [24,25,26]. Moreover, the pandemic, which was a major global environmental event, offers a unique opportunity to review in-person teaching from a public health perspective and raise the possibility of changing school schedules or using widely available online tools to improve the health and daily lives of adolescents.

We report data of a cross-sectional survey conducted at three competitive high schools in Vilnius, Lithuania. The aim of our study was to assess how high school students perceive their (1) sleep, (2) mental health, (3) physical health and (4) learning experience during COVID-19 lockdowns and after school reopening.

We intended to test the following hypotheses:

**H_1_.** 
*High school students perceive their sleep quality to be better and sleep duration to be longer during school closure.*


**H_2_.** 
*High school students report worse mental and physical health during lockdowns.*


**H_3_.** 
*Online learning is perceived to be of lower quality than learning in person.*


**H_4_.** 
*After school reopening, the relationship between quality of sleep and the perceived quality of studies is mediated by symptoms of anxiety and depression.*


## 2. Materials and Methods

### 2.1. Subjects and Procedure

An online anonymous questionnaire was distributed through closed communication groups among students of three high schools (9th to 12th grade) in Vilnius, the capital of Lithuania, from 20 October 2021 to 20 November 2021. High schools were selected based on their academic performance in the academic year 2020/2021—a non-random convenience sample was sought from the three best performing public schools in the city. The inclusion criteria were being enrolled in one of the three schools and having completed and submitted answers to each item of the questionnaire. Only responses to open-ended survey items judged to be intentionally misleading (e.g., 20 h of commute per day, 30,000 min of being outside daily, etc.) were excluded from the analysis.

### 2.2. Measures

The questionnaire consisted of ten parts:Demographic data (age, sex, grade);Items rated on a scale from 0 to 10: (a) the quality of studies, (b) psychological health, (c) physical health, (d) sleep quality, (e) overall well-being (exhausted vs. well-rested) either at the time of the survey (i.e., while learning in person) or during studies online (i.e., retrospectively);The average of (a) hours spent studying, (b) hours of sleep, (c) minutes spent outside and (d) minutes of physical activity (including fast-paced walking) per day either at the time of the survey or during studies online;The average hours of commute while learning in person;The perceived advantages and disadvantages of online learning (multi-choice);COVID-19-related questions: being diagnosed with COVID-19 by polymerase chain reaction (PCR), being in isolation, using masks in the classroom, being vaccinated;Causes of pandemic-associated anxiety (multi-choice);The Beck depression inventory (BDI, scores from 0 to 63 with higher score indicating more expressed symptoms of depression). A cut-off score of ≥16 for the BDI has been proposed to differentiate between adolescents with and without relevant symptoms of depression (73.0% sensitivity, 80.3% specificity) [27,28]. The Cronbach α of the scale was 0.915;The Generalized anxiety scale-7 (GAD-7, scores from 0 to 21, with a higher score indicating more expressed symptoms of anxiety). A cut-off score of ≥8 for the GAD-7 provides 77% sensitivity and 82% specificity for detecting any anxiety disorder [29,30]. The Cronbach α of the scale was 0.914;The Pittsburgh sleep quality index (PSQI, scores from 0 to 21, with a higher score indicating worse sleep quality). A cut-off score of >5 for the PSQI provides 89.6% sensitivity and 86.5% specificity to detect poor sleep quality [31]. The Cronbach α of the scale was 0.708.

### 2.3. Statistical Analysis

A sample of at least 172 was needed to include up to 10 independent variables in a multiple linear regression model with α = 0.05, 1 − β = 0.95 and f^2^ = 0.15 (computed by entering respective input parameters in G*Power 3.1.9.7). Data analysis was carried out in IBM SPSS Statistics 26 (IBM, Chicago, IL, USA). The Kolmogorov–Smirnov test was used to assess the normality of the continuous variables. The Wilcoxon test was applied to compare two paired groups for variables measured at two time points: (1) at the time of the survey (i.e., while learning in person) and (2) during studies online (i.e., indicated retrospectively). The Kruskal–Wallis H and the Mann–Whitney U tests were employed for comparisons of independent subgroups. Variables associated with BDI, GAD-7 and PSQI were investigated through Spearman’s correlations and by performing a stepwise linear regression analysis with PSQI as the dependent variable. A path analysis was conducted in Amos Graphics 22 to define the mediating role of anxiety and depression in the relationship between the perceived quality of studies (while learning in person) and the quality of sleep.

## 3. Results

There were 643 responses, of which 628 (97.7%) comprised the final study sample (individuals who provided inappropriate and/or implausible answers were excluded). The main characteristics of the sample are presented in Table 1. There were 268 (42.7%) individuals who scored ≥16 on the BDI, 342 (54.5%) who scored ≥8 on the GAD-7 and 398 (63.4%) who scored >5 on the PSQI.

On average, students perceived the quality of their studies and their physical health to be better while studying in person (Table 2). Students of the 10th grade reported the worst study quality (Kruskal–Wallis H(3) = 15.24, *p* = 0.002) and psychological health (H(3) = 17.62, *p* = 0.001) while learning in person. Final-year students had worse physical health (H(3) = 15.00, *p* = 0.002), slept less (H(3) = 10.07, *p* = 0.018), spent less time outside (H(3) = 12.21, *p* = 0.007) or engaging in physical activities (H(3) = 22.99, *p* < 0.001) than younger adolescents.

Respondents of the 10th grade were more favorable towards the quality of studies online (H(3) = 25.98, *p* < 0.001), reported better psychological as well as physical health during lockdowns (H(3) = 9.13, *p* = 0.028 and H(3) = 10.79, *p* = 0.013, accordingly). They also had better sleep (H(3) = 14.24, *p* = 0.003), reported higher levels of energy (H(3) = 9.73, *p* = 0.021) and exercised more (H(3) = 8.30, *p* = 0.040) than 12th-year students while studying online.

Female students had less favorable views towards online (Z = −2.86, *p* = 0.004) but not in-person (Z = −0.68, *p* = 0.497) learning. They also regarded their physical and mental health as well as their sleep quality and well-being to be worse both while learning online and in-person (*p* < 0.05). Female respondents scored higher on the BDI (Mdn = 16 vs. Mdn = 9, Z = −8.08, *p* < 0.001), GAD-7 (Mdn = 10 vs. Mdn = 5, Z = −8.11, *p* < 0.001) and PSQI (Mdn = 7 vs. Mdn = 5.5, Z = −5.57, *p* < 0.001) as well. There was no difference in these measures between individuals who reported COVID-19 infections and the ones who did not.

Opinions about studying online and reported COVID-19-related problems are presented in Table 3.

Measures of anxiety (r_s_ = 0.57, *p* < 0.001) and depression (r_s_ = 0.62, *p* < 0.001) were correlated with sleep quality (Figure 1) and associated with subjective estimates of health and daily-life-related aspects during studies in person (Table 4 and Table 5).

A mediation analysis revealed that the association between the perceived quality of studies and sleep quality (standardized direct effect −0.23, *p* < 0.001) is mediated by indirect effects of anxiety and depression (combined standardized indirect effect: −0.19, *p* < 0.001), Figure 2.

## 4. Discussion

### 4.1. General Findings

We report the results of an online cross-sectional survey involving 628 high school students in Lithuania. The study sample was predominantly female; however, it had an even distribution of respondents from the four grades (9th to 12th). Students also had relatively high vaccination rates, and the vast majority were able to keep studying in person one month before completing the survey. It is of note that vaccination for COVID-19 was widely available in the country and alternative measures of infection control (e.g., rapid antigen or pooled PCR testing) were installed in schools to complement mask use, disinfection, social distancing and ventilation. Overall, we were able to fully confirm three of our hypotheses: high school students reported better and longer sleep during lockdown (H_1_); they perceived online learning to be of inferior quality as compared to learning in person (H_3_); and the relationship between the quality of sleep and the perceived quality of studies was mediated by symptoms anxiety and depression after school reopening (H_4_). Moreover, while high school students reported worse physical health during lockdowns, their evaluation of mental health was higher during school closures, thus only partially confirming H_2_. In summary, the survey confirmed previous findings that school reopening leads to decreased sleep duration and increased physical activity (i.e., changes opposite to those observed after the initiation of lockdowns) but did not support the alleviation of mental health issues (measured as a self-rated mental health status) post-lockdown [18,20]. To the best of our knowledge, the study is also the first to address the relationship between the quality of learning, anxiety, depression and the quality of sleep after school reopening.

### 4.2. Differences in Health and Quality of Learning during Online and In-Person Studies

Before the time of our study, Lithuanian students had experienced two simultaneous COVID-19-associated lockdowns and school closures (March–June 2020 and November 2020–June 2021) and returned to learn in person in September 2021 [32]. After school reopening in September 2021, participants of our study reported improved quality of their studies as well as physical health and spent more time studying or being physically active. On the other hand, sleep quality, sleep hours, energy levels and mental health were subjectively evaluated to be worse. Such results suggest that a “trade-off” may exist between (a) good academic experience and physical activity (learning in person) and (b) optimal sleep quality and good mental health (learning online). However, our data does not indicate that the two groups of variables (a and b) are mutually exclusive. For instance, correlation analysis provided evidence that well-evaluated studies do not preclude good sleep quality and low levels of depression and anxiety: in fact, the more students appreciated their studies in person, the less they scored on the PSQI, BDI and GAD-7. A path analysis suggested that better study quality is also related to better sleep through a negative association with both the BDI and the GAD-7. Further, more hours of physical activity and better self-rated physical health were both associated with better sleep quality and lower levels of depression and anxiety. Therefore, bad sleep quality and persisting mental health issues after school reopening are most likely determined by other factors than improved study quality and increased physical activity. Among them could be changes in school schedules (e.g., the need to wake up earlier to attend classes) and daily commutes, which may decrease sleep hours and be detrimental for students whose endogenous circadian cycles trend towards later awakening [13,14,15,33]. Longer study hours and the continuous risk of COVID-19 may increase anxiety. It is noteworthy that anxiety, depression and bad sleep may also have a bidirectional interaction [34]. While the cross-sectional design of our study does not allow direct comparison of depression or anxiety levels and sleep quality during and after lockdown, two in five students could be suspected to have a significant depressive disorder, more than half to have an anxiety disorder, while almost two-thirds surpassed the PSQI threshold for poor sleep at the time of the survey (i.e., while learning in person). Further studies are required, however, to reveal adjustments (e.g., later school start times, student counseling) that could be beneficial in improving students’ sleep as well as mental health after returning to school.

### 4.3. The Possible Influence of Demographic Factors on Unfavourable Views of Online Learning 

In our study, students of the 10th grade had a more positive outlook towards online learning. They represent individuals who were in lower secondary education (i.e., the 8th grade) at the start of the pandemic and competed to enter high schools during the lockdown in 2020. Because of the public health crisis, students that year were selected based on previous merit rather than results of entrance exams. During our study, adolescents of the 10th grade were in the last year of a general curriculum and had not yet chosen study streams for final-exam-oriented studies of the 11th and 12th years of school. Noticeably, male students were also more satisfied with online learning and had better sleep and fewer symptoms of anxiety and depression after school reopening. While the reasons underlying a better experience of learning online among 10th grade or male students remain unclear, our data indicate that high school students should not be regarded as a homogenous group, as some may benefit from online learning more than others. On average, however, students were aware that the quality of their studies worsened during national lockdowns. It is noteworthy that our survey was based in some of the best high schools in the country and the pandemic’s impact on learning and gain of new knowledge was the most frequent cause for respondents’ anxiety. Therefore, the critical evaluation of online teaching may be explained by both high students’ expectations and the professors’ unpreparedness to rapidly transfer and adapt the curriculum for virtual teaching. It can be argued that the online form of teaching itself does not compromise the quality of education as long as students’ technological needs are met and studying is made a social and engaging process [35]. However, more than half of all students missed interactions with their peers or tutors and reported insufficient assistance from the teachers during virtual studies. It is thus important to remember that schools are also valuable social environments and should welcome students back as soon as adequate safety measures are available [6,7,36].

### 4.4. Practical Applications

Practical implications of our study concern the applicability of online learning in high schools as well as possible modifications to the current framework of studying in person. The results of our survey support the argument that lockdowns, which mostly simulated a situation of later school start times, were beneficial for the sleep of high school students [16]. It should also be noted that novel school policies (e.g., later school start times, the introduction of online courses) implemented specifically for students to sleep longer could have positive consequences for their mental and physical well-being [37]. However, high school students in our study were also aware of the substandard quality of their studies during lockdowns. As our findings indicate, the reduction in the quality of studies may itself be associated with symptoms of anxiety and depression or poor sleep. Thus, the complex relationship between mental health, physical well-being and learning should be acknowledged before implementing any new policies that could risk undermining high academic standards.

### 4.5. Limitations and Future Directions

Limitations of the current study are largely determined by its cross-sectional design: longitudinal data are needed to better define factors associated with poor sleep and mental health issues among high school students throughout the pandemic. Further, our survey was based in highly ranked schools of secondary education in a high-income country in Europe; thus, the results presented cannot be generalized to student populations of dissimilar social and economic backgrounds. Participation bias may have influenced results as students with no interest in their health or study quality were possibly less likely to participate. Finally, the current study relied only on subjective self-reported information and objective methods of data collection, such as actigraphy to determine sleep-related variables and structured interviews to investigate mental health problems, are needed to confirm our findings.

Future controlled trials are also required to better define the psychological and physical outcomes among students learning online (or in a hybrid way) compared to those attending live classes. In our study, only estimates of self-rated mental and physical health were compared while studying online during lockdowns versus learning in person after school reopening. Single-item self-reported measures are increasingly appreciated as tools that reflect future morbidity; however, they do not address the presence of distinct treatable conditions (e.g., depression) [25,26]. It is therefore important that the quality of sleep and symptoms of anxiety and depression are investigated in the setting of teaching online (or in a hybrid way) after the COVID-19 pandemic. Such studies could help determine, what are the positive and negative health effects of online learning in high schools in the absence of a global pandemic.

## 5. Conclusions

The results of the current study suggest that the reopening of high schools during the COVID-19 pandemic was beneficial for the quality of teaching and students’ physical health but had negative effects on their sleep and mental health. Around half of the high school students reported significant levels of anxiety, while almost two-thirds reported having poor sleep. Online learning was perceived to be of worse quality than studying in person, and problems gaining new knowledge were the primary cause for anxiety. Poor study quality after school reopening was related to worse quality of sleep both directly and through symptoms of anxiety and depression. In summary, our findings indicate that high school students could potentially benefit from new school policies aimed to improve student sleep while respecting high educational standards. Future studies should investigate options to improve sleep and mental health among high school students while ensuring study quality and students’ safety.

## Figures and Tables

**Figure 1 ijerph-19-02553-f001:**
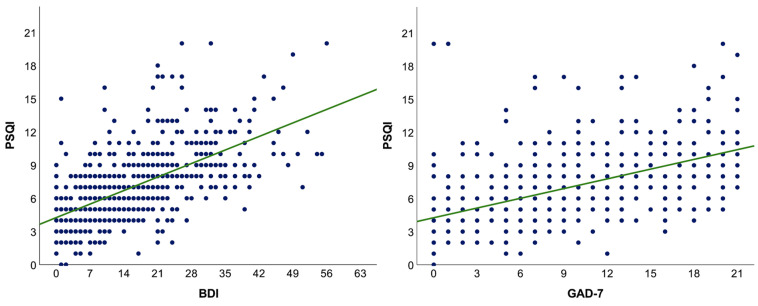
The relationship between sleep quality and symptoms of depression or anxiety. A linear regression fit line is presented in green. BDI—the Beck depression inventory; GAD-7—the Generalized anxiety scale-7; PSQI—The Pittsburgh sleep quality index.

**Figure 2 ijerph-19-02553-f002:**
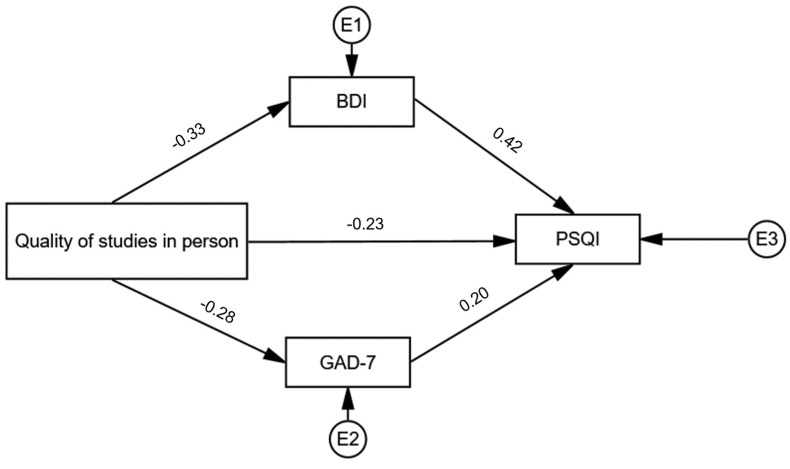
A mediation analysis of the relationship between the reported quality of studies and the quality of sleep (PSQI) through symptoms of anxiety (GAD-7) and depression (BDI). Standardized effect estimates are presented. BDI—the Beck depression inventory; E—error term; GAD-7—the Generalized anxiety scale-7; PSQI—The Pittsburgh sleep quality index.

**Table 1 ijerph-19-02553-t001:** General characteristics of the participants in the study.

Characteristic		n (%) or Mean (SD)
Number of respondents		628
Sex	Female	442 (70.4)
	Male	186 (29.6)
Age		16.1 (1.2)
Grade	9th	165 (26.3)
	10th	198 (31.5)
	11th	151 (24.0)
	12th	114 (18.2)
COVID-19 PCR+		69 (11.0)
COVID-19 Hospitalization		2 (0.3)
Wears masks at school		627 (99.8)
Class had to isolate since September 2021		39 (6.2)
Days in isolation		0.6 (2.4)
In isolation during survey		6 (1.0)
Had only classes in person during the past month		563 (89.6)
Vaccinated against COVID-19		530 (84.4)
Parents agreed (or would agree) with COVID-19 vaccination		569 (90.6)
BDI		15.4 (11.3)
GAD-7		9.2 (5.9)
PSQI		6.9 (3.3)

BDI—the Beck depression inventory; GAD-7—the Generalized anxiety scale-7; PCR+—a positive polymerase chain reaction test; PSQI—The Pittsburgh sleep quality index.

**Table 2 ijerph-19-02553-t002:** A comparison of health and daily-life-related factors during classes online and in-person.

Characteristic	Classes Online	Classes in Person	Wilcoxon Test Statistic
Quality of studies ^a^	7 (0–10)	8 (0–10)	−12.435 ***
Mental health ^a^	7 (0–10)	6 (0–10)	−2.220 *
Physical health ^a^	7 (0–10)	8 (0–10)	−9.176 ***
Sleep quality ^a^	9 (0–10)	5 (0–10)	−19.489 ***
Well-being (exhausted vs. well-rested) ^a^	8 (0–10)	4 (0–10)	−14.733 ***
Hours of studying per day ^b^	7.5 (2.7)	9.2 (2.6)	−15.580 ***
Hours of sleep per day ^b^	8.1 (1.2)	6.2 (1.2)	−19.509 ***
Hours of commute per day ^b^	n/a	1.7 (0.9)	n/a
Minutes outside per day ^b^	63.5 (62.0)	82.4 (67.8)	−6.848 ***
Minutes of physical activity per day ^b^	53.3 (55.7)	63.0 (62.7)	−4.344 ***

a—value on a scale from 0 (worst) to 10 (best); median (range); b—mean (standard deviation); n/a—not applicable; *—*p* < 0.05; ***—*p* < 0.001.

**Table 3 ijerph-19-02553-t003:** The outlook of high school students towards online learning and the COVID-19 pandemic.

Characteristic	n	%
**Advantages of online learning**		
More time to sleep	557	88.7%
Better teaching quality	54	8.6%
It is safer during the pandemic	469	74.7%
More free time	422	67.2%
Easier to keep a good diet	308	49.0%
No need to wear a mask	464	73.9%
No need to commute	454	72.3%
A more comfortable learning environment	299	47.6%
**Disadvantages of online learning**		
Impossible to meet friends	411	65.4%
Impossible to attend extracurricular activities	168	26.8%
Lack of help from teachers	325	51.8%
Lack of interaction with teachers and school staff	319	50.8%
Lack of physical activity	273	43.5%
Worse teaching quality	401	63.9%
A less comfortable learning environment	160	25.5%
**Causes for anxiety: the impact of the pandemic on....**		
learning and gain of new knowledge	385	61.3%
my family’s health	277	44.1%
my health	264	42.0%
future studies and career	260	41.4%
my relationships with friends	235	37.4%
my relationship with family members	88	14.0%
the social and economic situation of my family	170	27.1%
**Reported causes for not vaccinating ^a^**		
Previous COVID-19 infection	37	37.8%
Because of possible side-effects	32	32.7%
My parents/guardians do not agree with me being vaccinated	31	31.6%
I do not believe the vaccine is effective to stop the spread of COVID-19	14	14.3%
I do not believe the vaccine is effective against COVID-19	13	13.3%
I cannot be vaccinated as confirmed by a physician	4	4.1%
**Student-reported effects of wearing face masks**		
Lack of air	268	42.7%
Headache	187	29.8%
Increased fatigue	256	40.8%
Skin problems	318	50.6%
No effects	154	24.5%

a—the percent of unvaccinated individuals is indicated.

**Table 4 ijerph-19-02553-t004:** Spearman’s correlations between health and daily-life-related factors while studying in person and scores of sleep quality as well as anxiety scales.

Characteristic (While Studying in Person)	PSQI	GAD-7	BDI
Quality of studies	−0.36 ***	−0.30 ***	−0.35 ***
Mental health	−0.40 ***	−0.47 ***	−0.53 ***
Physical health	−0.36 ***	−0.31 ***	−0.41 ***
Sleep quality	−0.56 ***	−0.38 ***	−0.38 ***
Well-being (exhausted vs. well-rested)	−0.54 ***	−0.51 ***	−0.54 ***
Hours while studying per day	0.24 ***	0.27 ***	−0.22 ***
Hours of sleep per day	−0.58 ***	−0.32 ***	−0.33 ***
Hours of commute per day	0.12 *	0.06	−0.07
Minutes outside per day	−0.09 *	−0.07	−0.09 *
Minutes of physical activity per day	−0.14 ***	−0.19 ***	−0.22 ***

*—*p* < 0.05; ***—*p* < 0.001; BDI—the Beck depression inventory; GAD-7—the Generalized anxiety scale-7; PSQI—The Pittsburgh sleep quality index.

**Table 5 ijerph-19-02553-t005:** Results of a stepwise linear regression model with PSQI as the dependent variable (F(7,620) = 113.97, *p* < 0.001, R^2^_ad_j = 0.558).

Independent Variables	β_standardized_	t	*p*
Constant	n/a	17.733	<0.001
BDI	0.331	8.309	<0.001
Hours of sleep per day	−0.282	−8.575	<0.001
Quality of sleep ^a^	−0.202	−5.700	<0.001
Quality of studies ^a^	−0.138	−4.490	<0.001
GAD-7	0.133	3.488	0.001
Mental health ^a^	0.095	2.709	0.007
Minutes outside per day	−0.060	−2.242	0.025

Excluded variables: grade, sex, COVID-19 (confirmed by PCR, yes/no), isolation during the survey (yes/no), physical health ^a^, well-being (exhausted vs. well-rested), hours of studying per day, hours of commute per day, minutes of physical activity per day, subjective evaluation from 0 (worst possible) to 10 (best possible) while studying in person; BDI—the Beck depression inventory; GAD-7—the Generalized anxiety scale-7; PSQI—The Pittsburgh sleep quality index; n/a—not applicable.

## Data Availability

Data supporting the findings may be provided from the corresponding author upon request.

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
