# Peer review of "Sleep Quality, Mental Health and Learning among High School Students after Reopening Schools during the COVID-19 Pandemic: Results of a Cross-Sectional Online Survey"

_ijerph, 2022, doi:10.3390/ijerph19052553_

Round 1

Reviewer 1 Report

Dear Editors,

Have a nice day.

This study compares the status of sleep quality, study quality, and mental health of high school students during virtual studies at home (due to COVID-19 lockdowns) and in-person studies after the reopening of schools. This is a nicely drafted research paper that meets the basic formalities of the introductory and methods part, however, the results part requires clarification. Please clarify from line 103 “(H(3)=15.24, p=0.002)” to line 111 what ‘H’ denotes here and why data given in these two paragraphs are not available in a tabulated form? Rest is fine from my side.  

Author Response

Dear Editors,

Have a nice day.

This study compares the status of sleep quality, study quality, and mental health of high school students during virtual studies at home (due to COVID-19 lockdowns) and in-person studies after the reopening of schools. This is a nicely drafted research paper that meets the basic formalities of the introductory and methods part, however, the results part requires clarification. Please clarify from line 103 “(H(3)=15.24, p=0.002)” to line 111 what ‘H’ denotes here and why data given in these two paragraphs are not available in a tabulated form? Rest is fine from my side.

Response

Have a nice day, 

We thank you for reviewing our manuscript and providing thoughtful comments. To answer your query regarding the “H”, please note that it refers to the reported statistic for the Kruskal-Wallis test (in the Methods section it was mentioned that “The Kruskal-Wallis H test was employed for comparisons of independent subgroups.”). For clarity, it is now specified at the first reported results that the Kruskal-Wallis test (otherwise known as the “H test”) was performed: “Kruskal-Wallis H(3)=15.24, p=0.002”. We kindly note that if we wanted to present the group comparisons of this part of the results in tabulated form, we would need a 4*19 table (4 grades*19 variables). Therefore, we believe it is more concise and convenient for the readers if we report statistically significant results in the two paragraphs.

Reviewer 2 Report

This is an important study as research has shown COVID-19 affects mental health. I believe this study adds to the body of literature in this field as research in high school students has been limited. My suggestions for the authors are below:

  1. add study setting and design to the title
  2. add most important findings with p values in abstract
  3. line 55 - describe the total number of items and questionnaire parts, not just describe them as several
  4. line 81 - explain sample size calculation
  5. line 102 - you write here that studying was better online, but in the abstract that it was better in person. I am confused, please explain
  6. line 142- start the discussion with the main findings

Author Response

This is an important study as research has shown COVID-19 affects mental health. I believe this study adds to the body of literature in this field as research in high school students has been limited.

Response:

We greatly appreciate your time and effort to review our manuscript. Please find point-by-point answers below.

My suggestions for the authors are below:

Point 1.  add study setting and design to the title

Response

We agree. The title now is “Sleep quality, mental health and learning among high school students after reopening schools during the COVID-19 pandemic: results of a cross-sectional online survey”

Point 2. add most important findings with p values in abstract

Response

We agree and have added the findings with p values to the abstract: “Students reported better study quality (Z=-12.435, p<0.001) and physical health (Z=-9.176, p<0.001), but worse sleep quality (Z=-19.489, p<0.001), shorter sleep duration (Z=-19.509, p<0.001) and worse self-reported mental health (Z=-2.220, p<0.05) while learning in person.”

Point 3. line 55 - describe the total number of items and questionnaire parts, not just describe them as several

Response

We agree. The number is now specified: “The questionnaire consisted of ten parts”

Point 4. line 81 - explain sample size calculation

Response

We now explain that the sample size was calculated in the program G*Power by entering the parameters mentioned before: “A sample of at least 172 was needed to include up to 10 independent variables in a multiple linear regression model with α=0.05, 1-β=0.95 and f2=0.15 (computed by entering respective input parameters in G*Power 3.1.9.7).”

Point 5. line 102 - you write here that studying was better online, but in the abstract that it was better in person. I am confused, please explain

Response

Thank you for noticing and pointing out the error. The sentence should be as follows: “On average, students perceived the quality of their studies and their physical health to be better while studying online in person”

Point 6. line 142- start the discussion with the main findings

Response

We agree and now start the discussion with the discussion of general findings and the hypotheses that were confirmed (please refer to “4.1. General findings”).

Reviewer 3 Report

Introduction.
1. there is a lack of description, explanation and operationalization of the psychological constructs adopted in the study (including mental health and others).
2. there is a lack of references to theories and concepts and mechanisms that form the theoretical background.

The aim of study
Hypotheses should be clearly detailed and described in this section.

Materials and Methods
1. A description of the reliability of the tools is missing. In reporting internal consistency reliabilities, please report the conventional Cronbach's alphas for all scales and subscales (possibly instead of the Omega coefficient).
2. A split should be made in the section, for example: 2.1. Subjects and Procedure, 2.2. Measures, 2.3. Statistical Analysis
3. Indicate the inclusion and exclusion criteria of study participants

Results
in my opinion, there have been simplistic statistical analyses applied. I would expect, based on theory and previous research, mediation analyses that account for covariates. You might think about using structural equation modeling or path analysis.

Discussion
Generally, the discussion section lacks theories and concepts references.
Moreover, there is no clear indication of which hypotheses have been confirmed and which have been rejected.
The discussion should be broadened, and the obtained research results should be related to specific theories and other research results in the studied area. In other words, the question arises: Which previous research results have been confirmed and which have not?.
I suggest dividing the section into practical implications, limitations, future research  and conclusions.
I suggest adding more detail and references to theories and research.
Concluding paragraph: should state clearly the main conclusions of the research and give a clear explanation of their importance and relevance.

Given the above objections, I regret to write that I think the article should be rejected.

Author Response

We thank the reviewer for the time and effort required to review our manuscript and provide us with insightful comments. We address each point raised below and would greatly appreciate reconsideration of the manuscript, which, we believe, has been improved during the revision stage.

Point 1. Introduction.

there is a lack of description, explanation and operationalization of the psychological constructs adopted in the study (including mental health and others).

Response

We agree that the introduction could benefit from a wider discussion considering the constructs that are later explored. We believe symptoms of anxiety and depression as well as measures of the quality of sleep to be well-known clinical concepts, thus, we used an additional paragraph of the introduction to emphasize our definition of health and, more specifically, mental health (concepts that are not clinimetrically defined in the Methods section):

“In recent years, the view of health being a state of “complete well-being” shifted towards a dynamic definition of health, which revolves around the capacity to “react to all kinds of environmental events having the desired emotional, cognitive, and behavioral responses and avoiding those undesirable ones” [22]. Similarly, recent conceptualizations of mental health emphasize that it is “a dynamic state of internal equilibrium which enables individuals to use their abilities in harmony with universal values of society” [23]. Self-rated single-item measures of overall mental health or the evaluation of scales used to detect selected clinical entities (e.g., anxiety, depression or poor quality of sleep) can help to quantify the benefits and disadvantages that the COVID-19 pandemic had for the health of the youngest generations of individuals in our societies [24–26]. Moreover, the pandemic, which was a major and global environmental event, offers a unique opportunity to review in-person teaching from a public health perspective and raise the possibility of changing school schedules or using widely available online tools to improve the health and daily lives of adolescents.”

Point 2. there is a lack of references to theories and concepts and mechanisms that form the theoretical background.

Response

We agree and have now provided references of discussions regarding the definition of health, the concept of mental health (as well as its measurement through single item scales), please refer to the comment above and references 22-26. The references for measures of anxiety, depression or poor sleep may be found in the Methods section (pre-existing scales and their association with clinical diagnostics of the disorders are presented).

Point 3. The aim of study

Hypotheses should be clearly detailed and described in this section.

Response

We agree. The following hypotheses are now presented:

“We intended to test the following hypotheses:

H1: High school students perceive their sleep quality to be better and sleep duration to be longer during school closure.

H2: High school students report worse mental and physical health during lockdown.

H3: Online learning is perceived to be of lower quality than learning in person.

H4: After school reopening, the relationship between quality of sleep and the perceived quality of studies is mediated by symptoms anxiety and depression.”

Point 4.  Materials and Methods

A description of the reliability of the tools is missing. In reporting internal consistency reliabilities, please report the conventional Cronbach's alphas for all scales and subscales (possibly instead of the Omega coefficient).

Response

The reliability of the scales used is now reported (0.915 for the BDI, 0.914 for the GAD-7 and 0.708 for PSQI). Please note that other items, such as self-reported mental health or study quality, were treated as individual Likert scale items and were not merged to form a novel scale.

Point 5. A split should be made in the section, for example: 2.1. Subjects and Procedure, 2.2. Measures, 2.3. Statistical Analysis

Response

We thank you for the suggestion. The suggested subheadings have now been added.

Point 6. Indicate the inclusion and exclusion criteria of study participants

Response

We have specified the inclusion/exclusion criteria applicable to our survey under 2.1. Subjects and Procedure: “The inclusion criteria were being enrolled in one of the three schools and having completed and submitted answers to each item of the questionnaire. Only responses to open-ended survey items judged to be intentionally misleading (e.g., 20 hours of commute per day, 30000 minutes of being outside daily etc.) were excluded from the analysis.”

Point 7 .  Results

in my opinion, there have been simplistic statistical analyses applied. I would expect, based on theory and previous research, mediation analyses that account for covariates. You might think about using structural equation modeling or path analysis.

Response

We agree with the suggestion to perform a path analysis. We based the latter on the result of linear regression and investigated how the perceived quality of studies are associated with PSQI (both directly and indirectly through symptomatology of anxiety and depression). The direct and indirect effects of quality of studies on sleep quality are reported (please also refer to Figure 2 for a visualization of the path analysis):

“A mediation analysis revealed that the association between the perceived quality of studies and sleep quality (standardized direct effect -0.23, p<0.001) is mediated by indirect effects of anxiety and depression (combined standardized indirect effect: -0.19, p<0.001), Figure 2.”

Point 8. Discussion

Generally, the discussion section lacks theories and concepts references.

Response

We appreciate your comment. Please refer to the answer to the point “I suggest adding more detail and references to theories and research.” below for changes we made in the Discussion.

Point 9.   Moreover, there is no clear indication of which hypotheses have been confirmed and which have been rejected.

Response

We now indicated which hypotheses have been confirmed under “4.1.General findings”:

“Overall, we were able to fully confirm three of our hypotheses: high school students reported better and longer sleep during lockdown (H1), they perceived online learning to be of inferior quality as compared to learning in person (H2), and the relationship between the quality of sleep and the perceived quality of studies was mediated by symptoms anxiety and depression after school reopening (H4). Moreover, while high school students reported worse physical health during lockdown, their evaluation of mental health was higher during school closures, thus only partially confirming H3.”

 Point 10.  The discussion should be broadened, and the obtained research results should be related to specific theories and other research results in the studied area. In other words, the question arises: Which previous research results have been confirmed and which have not?.

Response

We thank you for raising this point. Please note that the topic addressed (learning, depression, anxiety and sleep after reopening of high schools during the COVID-19 pandemic) is relatively novel and there remains little data to compare our findings with. However, we now mention which previous research results have been confirmed and which have not:

“In summary, the survey confirmed previous findings that school reopening leads to decreased sleep duration and increased physical activity (i.e., changes opposite to those observed after the initiation of lockdowns) but did not support alleviation of mental health issues (measured as a self-rated mental health status) post-lockdown [18,20]. To the best of our knowledge, the study is also the first to address the relationship between the quality of learning, anxiety, depression and the quality of sleep after school reopening.”

 Point 11.  I suggest dividing the section into practical implications, limitations, future research and conclusions.

Response

Thank you for the suggestion. We have introduced the subsections “Practical applications” and “Limitations and future directions” in the discussion (please note that “Conclusions” are present as well, as a separate section of the manuscript).

 Point 12.  I suggest adding more detail and references to theories and research.

Response

We thank you for the suggestion. We believe to provide references each time studies or theories by other study groups are being discussed. References regarding the definition of mental health has also been provided in the Introduction. Further, a broader discussion of the self-rating of mental health (i.e., what underlies this construct, references 25, 26) has been added with respective references as well. We believe our study was able to report clinimetric data only for the period of school reopening. We therefore emphasize under “4.4. Limitations and future directions” that future studies should investigate distinct clinical constructs rather than self-reported health (measured through self-reported single items):

“Future controlled trials are also required to better define the psychological and physical outcomes among students learning online (or in a hybrid way) as compared to those attending live classes. In our study, only estimates of self-rated mental and physical health were compared while studying online during lockdown versus learning in-person after school reopening. Single item self-reported measures are increasingly appreciated as tools that reflect future morbidity, however, they do not address the presence of distinct treatable conditions (e.g., depression) [25,26]. It is therefore important that the quality of sleep and symptoms of anxiety and depression are investigated in the setting of teaching online (or in a hybrid way) after the COVID-19 pandemic. Such studies could help determine, what are the positive and negative health effects of online learning in high schools in the absence of a global pandemic.”

Point 13.  Concluding paragraph: should state clearly the main conclusions of the research and give a clear explanation of their importance and relevance.

Response

We have now revised the concluding paragraph and have included a sentence to better outline the main idea of how our findings could be relevant for high schools:

“The results of the current study suggest that the reopening of high schools during the COVID-19 pandemic was beneficial for the quality of teaching and students’ physical health but had negative effects on their sleep and mental health. Around half of high school students reported significant levels of anxiety, while almost two thirds may be said to have poor sleep. Online learning was perceived to be of worse quality than studying in person and problems gaining new knowledge were the primary cause for anxiety. Poor study quality after school reopening was related to worse quality of sleep both directly and through symptoms of anxiety and depression. In summary, our findings indicate that high school students could benefit from new school policies aimed to improve student sleep while respecting high educational standards. Future studies should investigate options to improve sleep and mental health among high school students while ensuring study quality and students’ safety.”

 Point 14.  Given the above objections, I regret to write that I think the article should be rejected.

Response

We appreciate your opinion, but also believe to have thoroughly revised and improved the manuscript, according to the comments provided.

Reviewer 4 Report

The authors conducted a questionnaire study on the sleep quality, mental health and learning for high school students before and after the school reopen. The paper is well-written and organized, and the conclusions are supported by the data analysis. I would recommend publication if the authors could address the following minor issue:

(1) If possible, I advice the authors to include previous studies during the beginning of the school shutdown for a comparison, as the answers to the questions may be biased due to the chronological order of the school closing/opening event

(2) The authors have sampled a lot of biographical features such as gender/age for each student, wouldn't it be more interesting if the authors could conduct analysis on those features and see if they have any combined effect with the school reopening on the sleep quality/mental health/learning?

Author Response

We greatly appreciate the comments of the reviewer and are thankful for considering our manuscript. Please find answers to the comments below.

 The authors conducted a questionnaire study on the sleep quality, mental health and learning for high school students before and after the school reopen. The paper is well-written and organized, and the conclusions are supported by the data analysis. I would recommend publication if the authors could address the following minor issue:

 Point 1.

  • If possible, I advice the authors to include previous studies during the beginning of the school shutdown for a comparison, as the answers to the questions may be biased due to the chronological order of the school closing/opening event

 Response

We agree that the timeline of school closures and reopening should be considered. We kindly note that in Lithuania all schools acted according to orders of the Ministry of Education, Science and Sport and, therefore, experienced the transitions simultaneously. We now specify this under 4.2: “Before the time of our study, Lithuanian students had experienced two simultaneous COVID-19-associated lockdowns and school closures”. While we have not found any prior studies investigating the population of high school students in Lithuania during the pandemic, we now report which findings relating to school reopening were confirmed (as compared to studies by other research groups):

“In summary, the survey confirmed previous findings that school reopening leads to decreased sleep duration and increased physical activity (i.e., changes opposite to those observed after the initiation of lockdowns) but did not support alleviation of mental health issues (measured as a self-rated mental health status) post-lockdown [18,20].”

Point 2.

  • The authors have sampled a lot of biographical features such as gender/age for each student, wouldn't it be more interesting if the authors could conduct analysis on those features and see if they have any combined effect with the school reopening on the sleep quality/mental health/learning?

 Response

We agree. While the measure of grade (9th to 12th) was chosen instead of age in our analysis (based on the presumption it better represents the social and educational environment), we certainly agree that sex differences (alongside differences of those who had COVID-19 or not) should be investigated. We now report such results (please refer to the paragraph below). Please also note that the influence of the students’ grade has already been reported (paragraphs below Table 1) and that both sex and grade were included in the list of variables for stepwise linear regression (however, they were excluded from the model because they did not improve the model fit – please refer to the footnote of Table 5).

Newly reported results:

“Female students had less favourable views towards online (Z=-2.86, p=0.004), but not in-person (Z=-0.68, p=0.497) learning. They also regarded their physical and mental health as well as their sleep quality and well-being to be worse both while learning online and in person (p<0.001). Female respondents scored higher on the BDI (Mdn=16 vs Mdn=9, Z=-8.08, p<0.001), GAD-7 (Mdn=10 vs Mdn=5, Z=-8.11, p<0.001) and PSQI (Mdn=7 vs Mdn=5.5, Z=-5.57, p<0.001) as well. There was no difference in these measures between individuals who reported COVID-19 infections and the ones who did not.”

Round 2

Reviewer 1 Report

I think the authors have clarified the raised concerns in the first draft. I am satisfied with the revised version for publication.  

Reviewer 3 Report

I have no comments. The authors followed all the directions and suggestions. The article can be published.